# Influence of Microwave Frequency and Gas Humidity on the In-Vitro Blood Coagulation in Cold Atmospheric Pressure Plasma

**Jie Yu, Li Wu * and Kama Huang**

IAEM (Institute of Applied ElectroMagnetics), College of Electronics and Information Engineering,
Sichuan University, Chengdu 610065, China; m18188344135@163.com (J.Y.); kmhuang@scu.edu.cn (K.H.)
* Correspondence: wuli1307@scu.edu.cn

**Abstract:** In this article, the effects of microwave frequency (2450 MHz and 5800 MHz) and gas humidity (1%, 2%, 3%, 4%, 6% and 8%) on in vitro blood coagulation with cold atmospheric pressure plasma (CAPP) were investigated. The generation of reactive oxygen species (ROS, OH, O) was measured by optical emission spectra. The exposure temperature on blood droplets under treatment was below 55 °C in all cases, to avoid the thermal effect of plasma on the blood clotting. Investigations showed that, with the increase of frequency, the doses of ROS increased, the blood sample presented a more serious collapse and its surface became drier. The humidity of ionized gas can also accelerate the generation of ROS and the process of blood clotting. Our results propose a method to accelerate in vitro blood coagulation in CAPP by adjusting microwave frequency and gas humidity, and suggest a clinical benefit for plasma treatment as a coagulation device in surgery.

**Keywords:** blood coagulation; cold atmospheric plasma; microwave frequency effect; gas humidity

## 1. Introduction

Plasma medicine has emerged as an independent discipline [1–3], which has brought about tremendous changes to the medical domain. Compared with conventional thermal plasma, cold atmospheric pressure plasma (CAPP) without high temperature does not cause heat-sensitive and unavoidable living tissue damage [4–6]. Some biomedical applications have emerged, for example, Hong et al. [7] presented a dielectric barrier discharge air-plasma driven by 60 Hz ac high-voltage power and its application in disinfecting *E. coli* cells. Kim et al. [8] reported a spray type plasma torch system with helium and oxygen working gases to study the invasion activity in colorectal cancer cells. Yan et al. [9] described a pulsed cold plasma-induced blood coagulation via direct contact. In these plasma generators the reactive oxygen species (ROS) have played a key role in biological applications. Especially, reactive atomic oxygen (RAO) and other likely ROS on the activation of erythrocyte–platelet particles have been considered as a plausible mechanism of blood coagulation [10–12]. In order to explore the factors that affect blood coagulation, Won et al. [13] compared the influence of discharge plasma driving frequency with 900 MHz and 20 kHz on blood coagulation. Their study demonstrated that the effect of 900 MHz microwave coagulation was twice as good as that of 20 kHz radio frequency. Recently, Wu et al. [14] investigated the discharge properties of a coaxial plasma jet at different microwave frequencies (433 MHz, 915 MHz, 2450 MHz, and 5800 MHz). It was found that the frequency of triggering plasma played a critical role in the plasma particle densities. In addition, previous investigations [15,16] certificated the ROS in plasma can be effectively influenced through increasing the gas humidity.

This paper investigated the effects of microwave frequency (2450 MHz and 5800 MHz) and different ratios of $H_2O/Ar$ mixed gas (1%, 2%, 3%, 4%, 6% and 8%) on the ROS and in vitro blood coagulation with a coaxial plasma source.

## 2. Experimental Setup

The discharge properties of CAPP driven by 2450 MHz under different gas humidities have been discussed in our previous study [14]. The device is handheld and designed based on a coaxial structure, as shown in Figure 1. Because of the nature of the coaxial transmission line, this generator also can be applied to 5800 MHz.

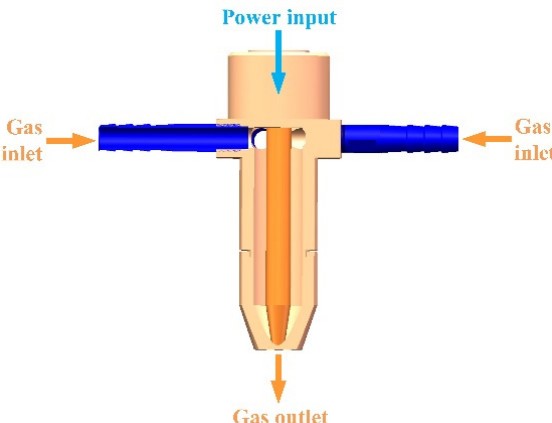

**Figure 1.** Schematic of CAPP device.

Figure 2 shows the schematic diagram of the plasma experimental system. This experimental system contains a solid-state source, a circulator, a dual-directional coupler, a matching load, a waveguide-to-coax converter and the plasma generator. The microwave power can be provided by specific frequency solid-state source, which is connected to a circulator and protected by it. A dual-directional coupler, linked to port 2 of circulator, is employed to measure the input and reflected powers with two power meters (AV2433, the 41st Institute of China Electronic Technology Group Corporation, Qingdao, China). The matching load is connected to the circulator port 3 for absorbing reflected power. The wave-to-coax converter is isolated to the other port of a dual-directional coupler to couple the microwave power to the coaxial structure. Argon with 1%, 2%, 3%, 4%, 6% and 8% humidity, its value is measured by humidity transmitter (LY60P-2X, ROTRONIC OEM, Bassersdorf, Switzerland), flows into the coax through four gas inlets. Its inflow rate is controlled by a flow controller (LF420-S, LAIFENG TECHNOLOGY CQLTD., Chengdu, China), which was adjusted to 8 L/min. The plasma emission spectra are measured by an optical emission spectrometer (AVSRACKMOUNT-USB2, Avantes, Apeldoorn, The Netherlands). Its optic probe is placed on the exit plane of the nozzle and 8 cm away from the nozzle horizontally. In this experiment, we set the input power to 20 W at frequency 2450 MHz and 5800 MHz.

The blood sample was exposed to CAPP, as presented in Figure 3. The movable holder can support specimens. All samples were placed on a glass slide with a distance of 1.5 cm directly below the plasma plume head. In the tests involving samples (10.0 μL), the exposure times of 30, 45, 60 and 75 s were chosen in these cases. During the exposure treatment, an optical fiber thermometer was employed during the experiments to monitor the temperature of samples constantly and make sure that they always were less than 55 °C.

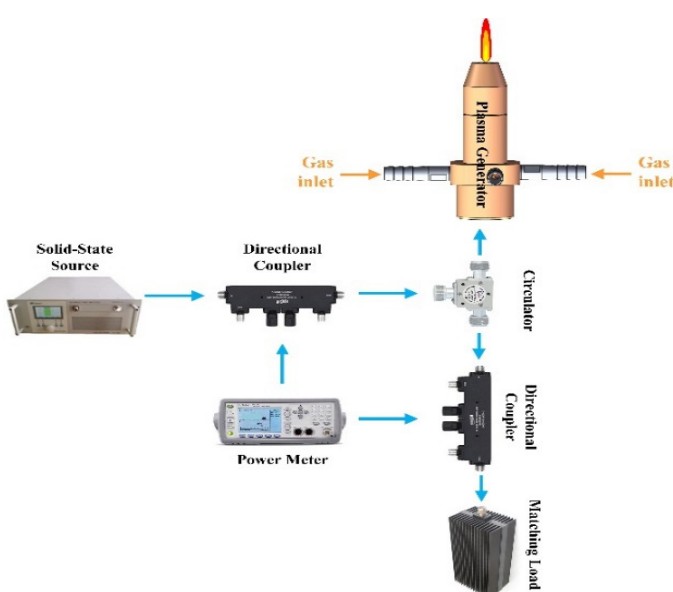

**Figure 2.** Experimental system schematic diagram.

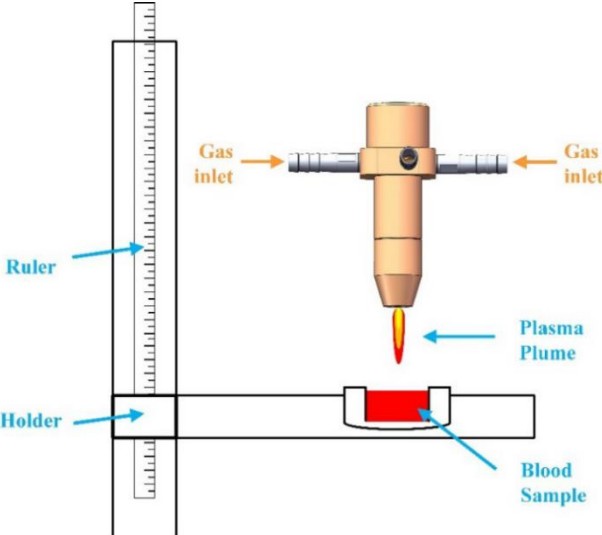

**Figure 3.** Schematic of CAPP treatment for a blood sample.

## 3. Materials and Methods

Blood samples, whole blood of sheep, were obtained from DOSSY EXPERIMENTAL ANIMALS CO., LTD, Chengdu, China. In order to prevent the blood from clotting prematurely, the blood was mixed with 3.2% sodium citrate solution in a ratio (volume) of 9:1. Sodium citrate is often used for anti-blood clotting in pharmaceutical industry [17]. All samples were divided into two main test groups. The first group was driven by two types of microwave, whose frequencies were 2450 MHz and 5800 MHz, including six samples exposed to different gas humidity plasma, respectively. In addition, the control group was to distinguish between the effects of heat and plasma on blood coagulation.

For further investigation, to explore the relationship between gas humidity and microwave frequency on blood clots, the emission spectra were used to record activated particles under different gas humidity and driving frequencies. In addition, the distributions of the electric field at the nozzle under the two frequencies were simulated.

## 4. Results and Discussion

### 4.1. Spectral Analysis

The emission spectra of Ar-H$_2$O mixture gas plasma plumes from 177 nm to 820 nm under the microwave input power of 20 W at 2450 MHz and 5800 MHz are shown in Figure 4. It can be observed that the emission spectra at these two frequencies were dominated by the Ar I line. Almost all the reactive radicals were generated richer in 5800 MHz microwave plasma, which means that the particles in 5800 MHz are more active than those in 2450 MHz. It is worth noting that RAO such as OH ($A^2 \overset{+}{\Sigma} - X^2 \prod$) at 309 nm and O ($3s^5\text{S} - 3p^5\text{P}$) at 777 nm exist in the plasma.

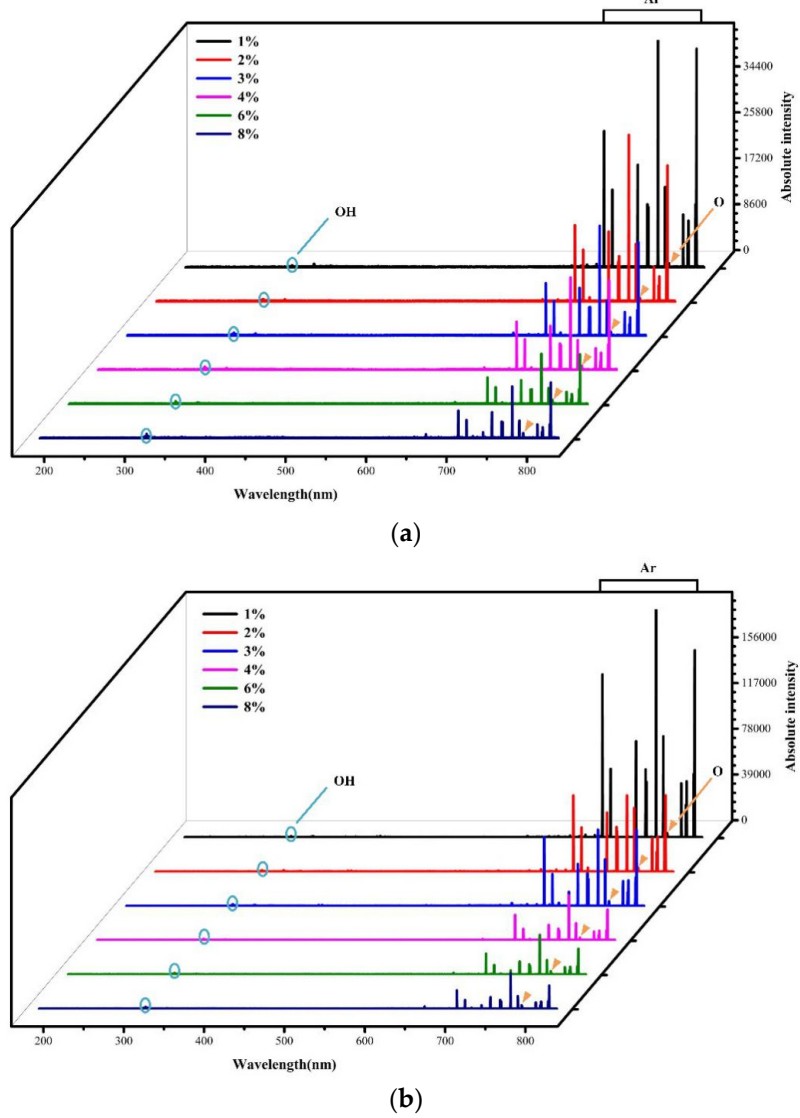

(**a**)

(**b**)

**Figure 4.** Spectra of plasma plumes under a microwave input power of 20 W (integration time of 30 ms) at different frequencies. (**a**) 2450 MHz, (**b**) 5800 MHz.

RAO and ROS have great potential in plasma medical treatments [3,18,19], such as killing the toughest biological agents [20], bacterial spores [21,22], affecting the formation of thrombosis in the vascular system [10], and so on. In our previous study [15], it was found that increasing the microwave input power and operating humidity helped to increase the doses of OH and O in the plasma. This work finds that the driving frequency also has impacts on the species density. The relationship of driving frequency and gas humidity on doses of OH and O at 2450 MHZ and 5800 MHz are described in Figure 5.

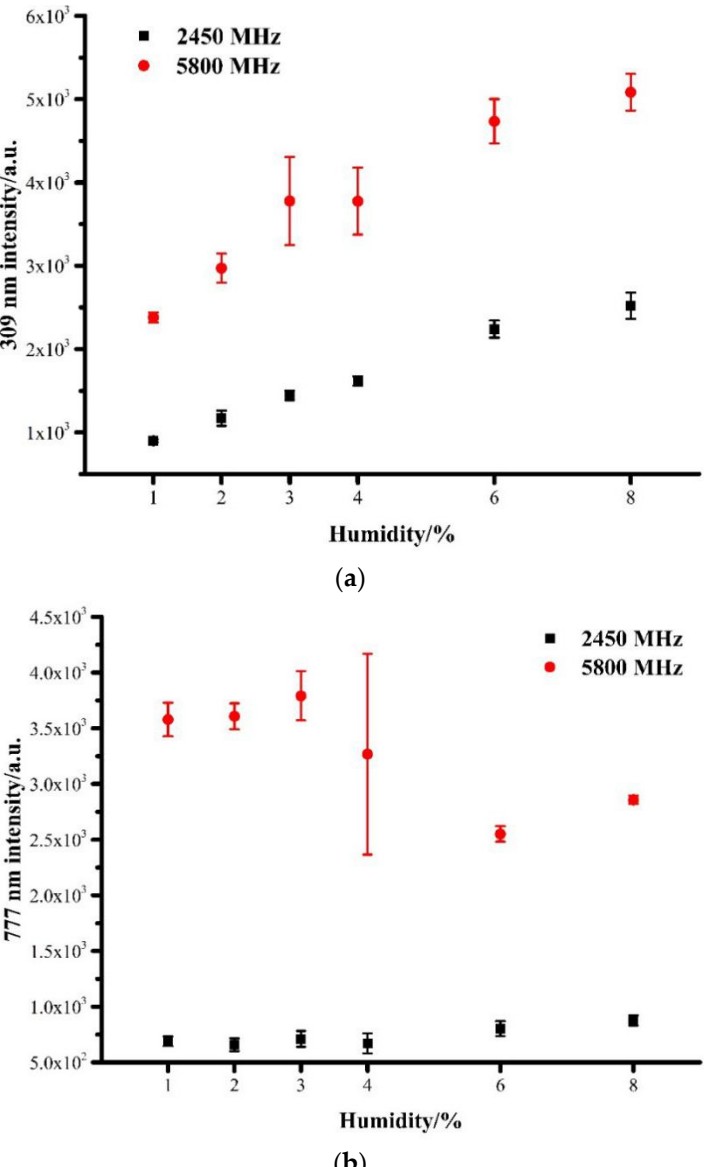

**Figure 5.** The emission spectra absolute intensity varies with humidity at different frequencies, (**a**) OH 309 nm line (integration time of 200 ms) and (**b**) O 777 nm line (integration time of 30 ms). Error bars represent the standard deviation of the measurements.

Figure 5a presents a visible, almost linear increasing of the OH relative line intensity with the humidity. However, the increasing of humidity does not always lead to higher densities of O. Instead, it shows a fluctuating trend (Figure 5b). This is because the strong electronegativity and large electron attachment surface of $H_2O$ seriously impeded the occurrence of collisions and formation of O when the water reached a threshold, which was expressed in our previous work [15]. It should also be noted that the microwave driving frequency had an impact on the doses of OH and O. The higher frequency, the more ROS.

In order to analyze why and how the frequencies affected the doses of ROS, the finite element method based software COMSOL and Maxwell equations were used to calculate the electric field distributions at these two frequencies with 20 W microwave input power. The calculated results are shown in Figure 6.

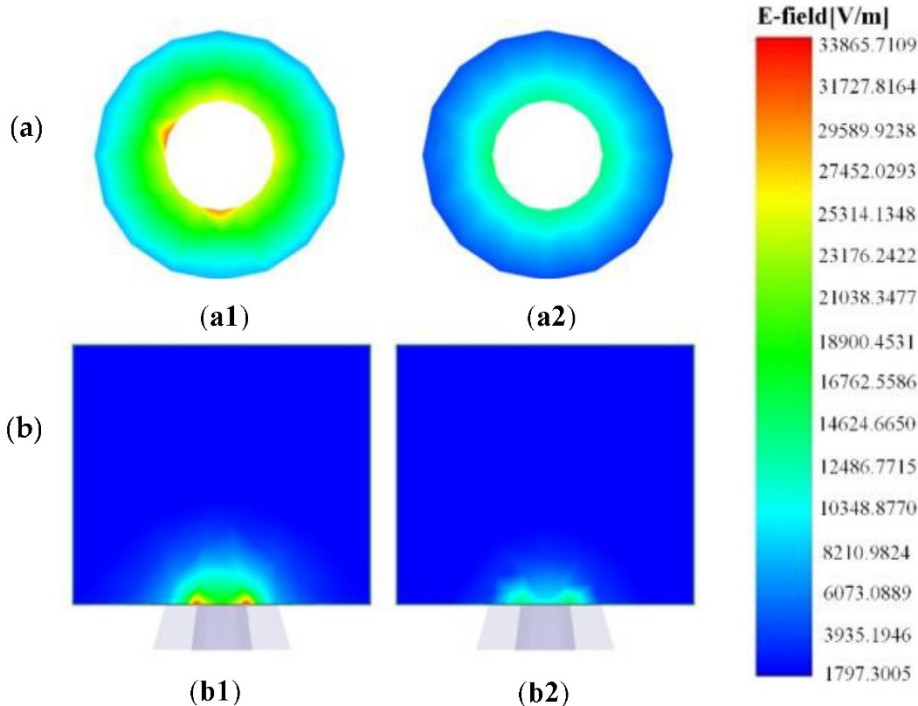

**Figure 6.** Simulated electric field intensity distributions at the nozzle (**a**), and side views of electric field distributions (**b**) at different operating frequencies and humidities. (**a1,b1**), 5800 MHz, (**a2,b2**), 2450 MHz, pure argon.

The calculated results (Figure 6) show that the electrical field intensity at 5800 MHz is much stronger than that at 2450 MHz. A higher electric field intensity can ionize more particles and produce a higher density plasma [14], which makes more particles collide with water molecules to generate more ROS.

### 4.2. Effect of Microwave Frequencies and Gas Humidies on Blood Coagulation

Figure 7 present the photos of two group samples treated with 2450 MHz and 5800 MHz plasma under different humidities (1%, 2%, 3%, 4%, 6%, and 8%) at an exposure distance of 15 mm. It reveals that the blood samples exhibit clot layer formation in the surface after exposing to plasma for 60 s or 75 s. The fluid exudation points covered on the surface of blood clot layer decreased and the blood samples were encrusted with an uneven shell. With the increase of gas humidity, more fluid exudation points were lost and the shell became rougher. This indicates that the plasma with higher gas humidity has a better potential in blood clotting. It is also worth noticing that the plasma with higher gas humidity required a shorter treatment time to form a clot layer on the blood samples.

Comparing Figure 7a,b, the shell of the former was smoother than that of the latter. The range of surface depressions increased with the rising frequency. The bubble of the shell in Figure 7b appeared collapsed. With the same exposure time, the higher the frequency, the more serious the collapse and the drier the surface of blood samples. After being treated for 75 s by the 8% gas humidity plasma at 5800 MHz, there was only a small spot left in the middle of the blood sample that had not yet coagulated.

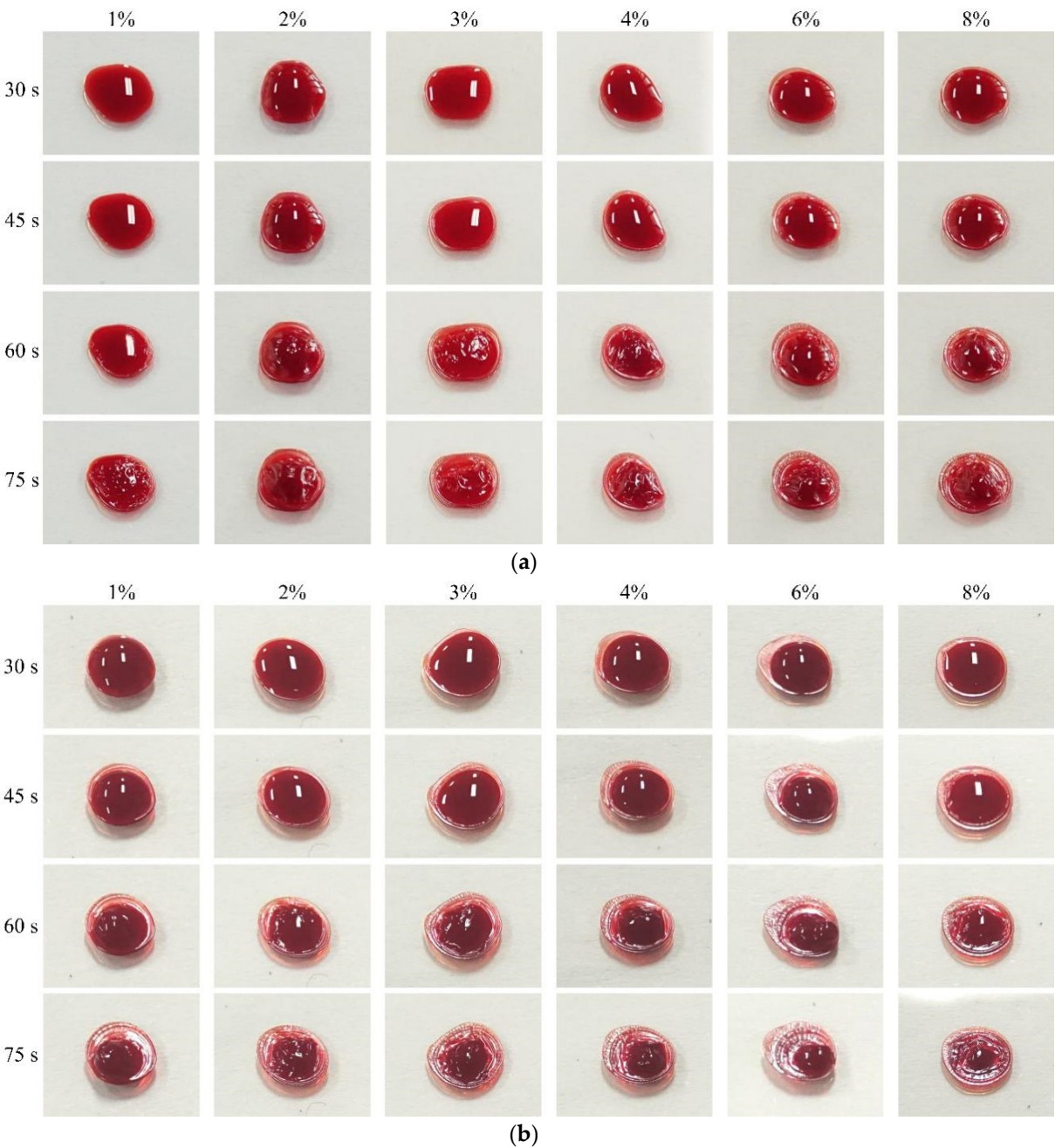

**Figure 7.** Blood samples treated by different microwave frequency plasma, (**a**) 2450 MHz, (**b**) 5800 MHz.

These phenomena should be attributed to the doses of ROS (reactive oxygen species and hydroxyl radicals). The higher the concentration, the more efficient the coagulation. Combined with the study of the coagulation mechanism of plasma, ROS causes blood clotting by inactivating the anticoagulation that controls clotting [23]. CAPP may accelerate the blood coagulation process by allowing the activation of platelets and promoting the formation of fibrin filaments in blood has been demonstrated in [9,24,25]. Therefore, the proposal to coagulate blood by CAPP with increased gas humidity to raise the doses of ROS can provide potential significance for biomedical applications, especially for early bleeding in trauma patients.

### 4.3. Treatment without and with Ar-H$_2$O Mixture Gas Plasma Jet

Although the discharge used here is essentially nonthermal, it may also transfer some heat energy to the sample. In order to separate the effect of heat and plasma on blood

clotting, the blood samples were exposed to a hot air blower for 30 s, 45 s, and 60 s within a treatment temperature of 55 °C. The treatment results are shown in Figure 8. In all cases, liquids are presented without a sign of coagulation.

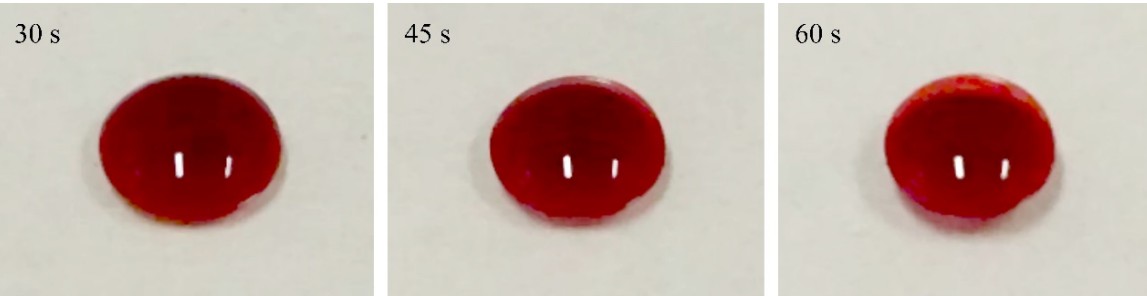

**Figure 8.** Blood samples treated by a hot air blower.

However, a shell, formed around the blood sample, as can be clearly seen in Figure 7a. These samples were treated with 4% humidity argon plasma at 2450 MHz microwave frequency with exposure time of 30 s, 45 s, and 60 s. This comparison shows that the presence of coagulation cannot be attributed to the heat effect.

## 5. Conclusions

The effects of microwave frequencies (2450 MHz and 5800 MHz) and ionized gas with various $H_2O/Ar$ ratios (1%, 2%, 3%, 4%, 6% and 8%) on the doses of ROS and the coagulation of anticoagulated blood in vitro by CAPP treatment have been experimentally studied. The optical emission spectra of plasma at the nozzle were detected by a spectrometer to reflect the doses of particles. This discharge seems to enhance the natural clotting process and promote rapid blood clotting. Our experiments have shown that:

1. The doses of ROS in plasmas detected are higher at 5800 MHz in the coaxial based CAPP. The humidity of ionized gas also promotes the generation of ROS.
2. Plasma with a higher gas humidity requires a shorter treatment time to form a clot layer on blood samples. Under 5800 MHz, the bubble of the shell appears more seriously collapsed.
3. The clotting effect in this work is attributed to the ROS in plasmas. Thermal effects are not the cause of clotting.

The paper reports our preliminary work about the influences of microwave frequency and gas humidity on in vitro blood coagulation. Quantitative measurement and comprehensive investigation on the mechanism of these two factors on blood coagulation will be conducted in the future.

**Author Contributions:** Conceptualization, L.W. and K.H.; methodology, J.Y.; data curation, J.Y. and L.W.; original draft preparation, J.Y.; review and editing, L.W. All authors have read and agreed to the published version of the manuscript.

**Funding:** This work was supported in part by the Science and Technology Planning Project of Sichuan Province under Grant 2018 HH0107. It is also supported in part by the National Natural Science Foundation of China under Grant 61801313 and Grant 61731013.

**Institutional Review Board Statement:** Not applicable.

**Informed Consent Statement:** Not applicable.

**Data Availability Statement:** The data that support the findings of this study are available from the corresponding author upon reasonable request.

**Conflicts of Interest:** The authors declare no conflict of interest.

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
