# Peer review of "Influence of Microwave Frequency and Gas Humidity on the In-Vitro Blood Coagulation in Cold Atmospheric Pressure Plasma"

_processes, doi:10.3390/pr9101837_

Round 1

Reviewer 1 Report

The presented investigation is important for the possible plasma applications in medicine and show an important dependance of the effects and ROS production with the electromagnetic field frequency. Usually much lower frequencies (e.g. 13.56MHz) are used for plasma sustaining while the higher frequencies shows higher efficiency and efficacy.

Some revision is needed before the paper is published.

In the title and everywhere in the text instead of "vitro" have to be written "in vitro" or "in-vitro".

The quality of the figures and the text inside them is not good and it is difficult to read and understand it.

p. 3 rows 85-88: The text is not clear. Language editing is needed. The authors have used "The first group was driven by two types 
microwave frequency with 2450 MHz and 5800 MHz..." which is not correct phrase. Similar phrase "two frequencies with 2450 MHz and 5800 MHz" is used on p. 7 row 133.
p. 6 rows 121-122 The authors concludes that "From these figures, it is obvious that the increasing of humidity does not leading to higher densities of OH and O. Instead, the ROS shows a fluctuating trend." This conclusion follows ONLY from Fig. 5(b) about O. In Fig. 5(a) is well visible increasing almost liner trend of the OH relative line intensity with the humidity.

Author Response

Response to Reviewer 1 Comments

Dear reviewer:

We are very grateful for your careful review regarding our manuscript entitled “Influence of microwave frequency and gas humidity on the vitro blood coagulation in cold atmospheric pressure plasma” (Manuscript # processes-1338471). We all agree that your comments are highly valuable and constructive for us. According to your comments, we have revised the manuscript thoroughly and carefully. We look forward to hearing from you regarding our submission. We would be glad to respond to any further questions and comments that you may have.

For all your concerns in your comments, below are our detailed responses. Hope both the revised manuscript and the responses to your comments can meet your expectations and requirements.

Thank you very much for your effort in reviewing our work again.

Best regards,

Jie YU, Li WU, and Kama HUANG

Point 1:The key term of "degree of blood clotting" introduced in Section 4.3, is not defined or made explicit in any way. Determining the blood clotting characteristics is not an ordinary task, existing various methods to do it in an objective manner (not based on observer assessment) and/or output results are expressed as numerical values. The paper cites several existing articles describing the effect of plasma on blood clotting which also include methods for evaluating this effect. Examples:

In [1] is presented (among other methods) the variation of Prothrombin concentration (mole/L) as a function of treatment time to evaluate plasma effect.

In [9] evaluation is based on transmission electron microscopy imaging.

In [13] is written:"....After treatment, the degree of coagulation was measured with capillary pipets (Hirschmann Laborgerate, ringcaps 100 uL). Coagulation rate was calculated from absorbed length ratio in the non-treated case. A 0 mm absorption length was considered to indicate 100% coagulation. Without plasma treatment, 20 uL of blood was absorbed by 1.07-cm-high along capillary pipets. Each result was an average of at least three measurements...."

None of these, or similar methods, have been used in paper.

Instead, two out of three conclusions (2 and 3) are based on results obtained by observing blood samples with the naked eye, before and after plasma treatment.

In my opinion, this is not a scientific approach.

Therefore, conclusions 2 and 3 of the paper are not based on measurable parameters.

Authors should use appropriate methods to compare their results with those of others, in order to prove the novelty of the work.

This as an important issue, which must be properly addressed.

Response 1: Thank you for your comments and suggestions. The phrase "degree of blood clotting" has been deleted in the whole manuscript.

There are various methods to investigate the effects of plasma on blood coagulation. References [1], [9] and [13] use objective manners like applying pipets or EMI to determine blood clotting characteristics. However, method of distinguishing the degree of blood coagulation by visual observation has also been used and reported in many papers.

In Reference [12], the author used plasma to treat nonanticoagulated whole blood and citrated whole blood. They reported that the blood sample exhibits an immediate clot layer formation on the surface exposed to plasma discharge by observer assessment.

Moreover, Ahmed K. et al.1, Janani E. et al.2 and Keying Z. et al.3 have used the method of visual observation to distinguish the blood clot degree. We believe that visual observation is helpful to determine if the blood sample could be coagulated by plasma or not. Therefore, we choose to keep the method and the corresponding results. To avoid the confusions and make it more appropriate, the conclusions have been revised.

The aims and novelty of this paper are to investigate the high frequency (up to 5800 MHz) and gas humidity on the ROS and blood coagulation. It is supposed to report our preliminary work about the influences of microwave frequency and gas humidity on in-vitro blood coagulation. Quantitative measurement of blood clotting degree and comprehensive investigation on the mechanism of these two factors on blood coagulation will be conducted in the future.

[1] Ahmed K M,  Eldeighdye S M,  Allam T M, et al. Power density measurements to optimize AC plasma jet operation in blood coagulation[J]. Australasian Physical & Engineering Sciences in Medicine, 2018, 41(3):1-12.

[2] Janani E,  Ghoranneviss M,  Al-Ebrahim M, et al. Blood Coagulation by Low Energy Plasma Jet[C]. ISPC, 2017.

[3] Keying Z, Qiliang C, et al. Effects of cold atmospheric plasma on coagulation function of rats [J]. Chinese Journal of Medical Physics, 2019, 036(003):351-355.

Point 2: At the end of Section 2 Experimental setup, there is the claim: "During the exposure treatment, an optical fiber thermometer can be used to monitor the temperature of samples, which are less than 55 C.".

It is an ambiguous statement.

It is not clear if either temperature has been presumed to be <55 C or it has been monitored constantly during the experiments (or in similar experiments, in similar conditions, etc) and always was <55 C. 

It must be clearly stated which is the case.

NB: "which is less than" or "which are less than"?

Response 2: Thank you for your comment. The temperatures of the samples were monitored constantly during the experiments and always were less than 55 C. It has been clearly stated in the revised manuscript and marked red.

Point 3: The section 4.1 Spectral Analysis, begins with:  "The emission spectrums of Ar-H2O mixture gas plasma plumes from 177 nm to 820 nm under the microwave input power of 20 W at 2450 MHz and 5800 MHz are shown in Fig. 4. It can be observed that the emission is dominated by the presence of Ar I line. Almost all the reactive radicals were generated richer in 5800 MHz microwave plasma, which means that the particles in 5800 MHz are more active than those in 2450 MHz."

In fact, the claim "Almost all the reactive radicals were generated richer in 5800 MHz microwave plasma.... " is convincingly proven, by the graphs shown in Fig 5(a), (b) rather than Fig. 4 (a), (b). 

Fig. 4 proves nothing in this matter.

The 309 nm line heights, at 2450 MHz and 5800 MHz, as shown in Fig. 4, are too small in both cases, being practically impossible to be visually compared. 

Moreover, because vertical scales of Fig 4 (a) and Fig 4 (b) are different, even Ar I lines, at 2450 MHz and 5800 MHz, are difficult to be visually compared.

My advice is to present Fig. 4 as it is, i. e. records of emission spectrum of Ar-H2O plasma plume demonstrating only that 309 nm and 777 nm lines exist.

Response 3: Thank you for your suggestion. We have deleted the comparison of the plasma spectrums at 2450 MHz and 5800 MHz, and revised the statements according to your suggestion.

Point 4: I think that correct term is "integration time", not "integral time", not "integrated time" in Fig. 4 caption and Fig. 5 caption.

Response 4: Thank you for your comment. It has been corrected in the revised manuscript.

Point 5: The software package and/or methods used to calculate the electric field intensity distribution shown in Fig. 6 must be specified.

Response 5: Thank you for your comment. The method used to calculate the electric field intensity distribution shown in Fig. 6 has been specified in the revised manuscript.

Point 6: As commented above: Conclusions 2 and 3 are not supported by quantitative measurements.

No numerical value is associated to the blood samples characteristics shown in Fig. 7, 8, 9.

Conclusions 2 and 3 are based only on human eyes visual assessment.

Most of the conclusion 3 (also inserted in Abstract), is trivial: "The coagulation effect is positively related to the exposure time and negatively related to the exposure distance."

Undoubtedly, at long distances and if the plasma is applied for a short time, the plasma effect is almost null.

No experiment is necessary for that.

Response 6: Thank you for your comment. These section about the exposure time and exposure distance of samples to the plasmas has been deleted in the revised manuscript.

Point 7: Partially, conclusion 1 as it is expressed: "The doses of ROS are positively associated with the increase frequencies.", could be false.

The ROS amount could increase with frequency because CAPP device (shown in Fig 1) simply performs better at 5800 MHz, due to its particular geometry. According to Fig. 6 electric field is higher at 5800 MHz compared to 2450 MHz, resulting, perhaps, in a more homogeneous and extended volume of plasma.

Response 7: Thank you for your comment. To avoid the confusions, conclusion 1 has been revised as “The doses of ROS in plasmas are detected higher at 5800 MHz in the coaxial based CAPP” in the revised manuscript. 

Point 8: Check the accuracy of the references list.

Several errors:

For [1] year is wrong (correct is 2008 not 2010)

For [3], volume, issue, are missing [9] and [26] are identical.

Response 8: Thank you for your comment. These the references have been corrected in the revised manuscript.

Point 9: Check English and terms used.

Some sentences are meaningless and some words are not appropriate.

For example (without being exhaustive):

In Abstract:

-"In this article, the effectiveness of.... The generation of ... were measured by optical emission spectra.", may be  "In this article, the effects of... The generation of ... was measured by optical emission spectra.", with "effects" instead of "effectiveness"  and  "was" instead of "were"?.

- What is "degree of clot" and how is it measured?

- What does  "....a prime importance of method...." mean?

In Section 1 Introduction:

-"This paper, in view of the researches before,....". 

What does "the researches before" mean? 

In Section 2 Experimental Setup:

-"Fig. 2 shows the two plasmas experimental system of schematic diagram".

What are "the two plasmas experimental system(s)"? In Fig. 2 is only one.

In Section 3 Materials and Methods:

-"To separate the effects of heat and plasma on blood coagulation, the second group was performed."

I do not understand the expression "the second group was performed".

In Section 4.1 Spectral Analysis:

-"In the past work, to raise the microwave input power and humidity of working gas can help to improve the doses of OH and O." -its meaning is difficult to understand.

In Section 4.2 Distributions of Electric Field Intensity:

-" In order to analysis why and how the frequencies affect the doses of ROS, the electric field distributions for two frequencies with 2450 MHz and 5800 MHz and two gas humidities with pure argon and 8% gas humidity at the nozzle of the plasma generator with 20 W of microwave input power were calculated and shown in Fig. 6." -its meaning is difficult to understand.

In Section 4.3 Effect of Microwave Frequencies on Blood Coagulation:

-"Fig. 7 present that the second group photos of the sample taken after treated by 2450 MHz and 5800 MHz plasma under different humidity (1%, 2%, 3%, 4%, 6%, and 8%) at an exposure distance of 15 mm."

 Expression: "second group photos of the sample" is unclear. May be:"photos of the second group of samples"?

In Section 4.4 Treatment without and with Ar-H2O Mixture Gas Plasma Jet:

-" The treatment results are shown in Fig. 8. In all cases, liquids are presented without a sigh of coagulation."

 "....a sigh of...." is probably ".... a sign of...." but is not a proper technical expression.

In section 5 Conclusions:

- "The optical emission spectra of plasma at the nozzle were detected by a spectrometer to reflect the doses of particles."

Text "to reflect the doses of particles" is not a proper technical expression.

- ".... The humidity of ionized gas also promotes the generated of ROS."

"generated of" instead of "generation of ".

Etc.

Response 9: Thank you for your comments and suggestions. We have revised the manuscript thoroughly and revised all the inappropriate parts in the revised paper, especially those in the abstract and the conclusions.

Reviewer 2 Report

Comments:

(1) The key term of "degree of blood clotting" introduced in Section 4.3, is not defined or made explicit in any way.

Determining the blood clotting characteristics is not an ordinary task, existing various methods to do it in an objective manner (not based on observer assessment) and/or output results are expressed as numerical values.

The paper cites several existing articles describing the effect of plasma on blood clotting which also include methods for evaluating this effect.
Examples:
In [1] is presented (among other methods) the variation of Prothrombin concentration (mole/L) as a function of treatment time to evaluate plasma effect 
In [9] evaluation is based on transmission electron microscopy imaging
In [13] is written:"....After treatment, the degree of coagulation was measured with capillary pipets (Hirschmann Laborgerate, ringcaps 100 uL). Coagulation rate was calculated from absorbed length ratio in the non-treated case. A 0 mm absorption length was considered to indicate 100% coagulation. Without plasma treatment, 20 uL of blood was absorbed by 1.07-cm-high along capillary pipets. Each result was an average of at least three measurements...."

None of these, or similar methods, have been used in paper. 

Instead, two out of three conclusions (2 and 3) are based on results obtained by observing blood samples with the naked eye, before and after plasma treatment.
In my opinion, this is not a scientific approach. 

Therefore, conclusions 2 and 3 of the paper are not based on measurable parameters.
Authors should use appropriate methods to compare their results with those of others, in order to prove the novelty of the work. 
This as an important issue, which must be properly addressed.

(2) At the end of Section 2 Experimental setup, there is the claim: "During the exposure treatment, an optical fiber thermometer can be used to monitor the temperature of samples, which are less than 55 C.". 
It is an ambiguous statement.
It is not clear if either temperature has been presumed to be <55 C or it has been monitored constantly during the experiments (or in similar experiments, in similar conditions, etc) and always was <55 C.  
It must be clearly stated which is the case. 

NB: "which is less than" or "which are less than"?.

(3)  The section 4.1 Spectral Analysis, begins with:  "The emission spectrums of Ar-H2O mixture gas plasma plumes from 177 nm to 820 nm under the microwave input power of 20 W at 2450 MHz and 5800 MHz are shown in Fig. 4. It can be observed that the emission is dominated by the presence of Ar I line. Almost all the reactive radicals were generated richer in 5800 MHz microwave plasma, which means that the particles in 5800 MHz are more active than those in 2450 MHz."

In fact, the claim "Almost all the reactive radicals were generated richer in 5800 MHz microwave plasma.... " is convincingly proven, by the graphs shown in Fig 5(a), (b) rather than Fig. 4 (a), (b).  
Fig. 4 proves nothing in this matter. 
The 309 nm line heights, at 2450 MHz and 5800 MHz, as shown in Fig. 4, are too small in both cases, being practically impossible to be visually compared.  

Moreover, because vertical scales of Fig 4 (a) and Fig 4 (b) are different, even Ar I lines, at 2450 MHz and 5800 MHz, are difficult to be visually compared.

My advice is to present Fig. 4 as it is, i. e. records of emission spectrum of Ar-H2O plasma plume demonstrating only that 309 nm and 777 nm lines exist. 

(4) I think that correct term is "integration time", not "integral time", not "integrated time" in Fig. 4 caption and Fig. 5 caption.

(5) The software package and/or methods used to calculate the electric field intensity distribution shown in Fig. 6 must be specified.

(6) As commented above: Conclusions 2 and 3 are not supported by quantitative measurements.
No numerical value is associated to the blood samples characteristics shown in Fig. 7, 8, 9.
Conclusions 2 and 3 are based only on human eyes visual assessment.

Most of the conclusion 3 (also inserted in Abstract), is trivial: "The coagulation effect is positively related to the exposure time and negatively related to the exposure distance."

Undoubtedly, at long distances and if the plasma is applied for a short time, the plasma effect is almost null. 
No experiment is necessary for that.

(7) Partially, conclusion 1 as it is expressed: "The doses of ROS are positively associated with the increase frequencies.", could be false.

The ROS amount could increase with frequency because CAPP device (shown in Fig 1) simply performs better at 5800 MHz, due to its particular geometry. According to Fig. 6 electric field is higher at 5800 MHz compared to 2450 MHz, resulting, perhaps, in a more homogeneous and extended volume of plasma.

(8) Check the accuracy of the references list. 
Several errors:
For [1] year is wrong (correct is 2008 not 2010)
For [3], volume, issue, are missing
[9] and [26] are identical.

(9) Check English and terms used.
Some sentences are meaningless and some words are not appropriate.
For example (without being exhaustive):

In Abstract:
-"In this article, the effectiveness of.... The generation of ... were measured by optical emission spectra.", may be  "In this article, the effects of... The generation of ... was measured by optical emission spectra.", with "effects" instead of "effectiveness"  and  "was" instead of "were"?.
- What is "degree of clot" and how is it measured?
- What does  "....a prime importance of method...." mean?

In Section 1 Introduction:
-"This paper, in view of the researches before,....".  
What does "the researches before" mean?  

In Section 2 Experimental Setup:
-"Fig. 2 shows the two plasmas experimental system of schematic diagram". 
What are "the two plasmas experimental system(s)"? In Fig. 2 is only one.

In Section 3 Materials and Methods:
-"To separate the effects of heat and plasma on blood coagulation, the second group was performed."
I do not understand the expression "the second group was performed".

In Section 4.1 Spectral Analysis:

-"In the past work, to raise the microwave input power and humidity of working gas can help to improve the doses of OH and O." -its meaning is difficult to understand.

In Section 4.2 Distributions of Electric Field Intensity:

-" In order to analysis why and how the frequencies affect the doses of ROS, the electric field distributions for two frequencies with 2450 MHz and 5800 MHz and two gas humidities with pure argon and 8% gas humidity at the nozzle of the plasma generator with 20 W of microwave input power were calculated and shown in Fig. 6." -its meaning is difficult to understand. 

In Section 4.3 Effect of Microwave Frequencies on Blood Coagulation:

-"Fig. 7 present that the second group photos of the sample taken after treated by 2450 MHz and 5800 MHz plasma under different humidity (1%, 2%, 3%, 4%, 6%, and 8%) at an exposure distance of 15 mm."

 Expression: "second group photos of the sample" is unclear. May be:"photos of the second group of samples"?

In Section 4.4 Treatment without and with Ar-H2O Mixture Gas Plasma Jet:
-" The treatment results are shown in Fig. 8. In all cases, liquids are presented without a sigh of coagulation."
 "....a sigh of...." is probably ".... a sign of...." but is not a proper technical expression.

In section 5 Conclusions:

- "The optical emission spectra of plasma at the nozzle were detected by a spectrometer to reflect the doses of particles."

Text "to reflect the doses of particles" is not a proper technical expression.

- ".... The humidity of ionized gas also promotes the generated of ROS."

"generated of" instead of "generation of ".

Etc.

Author Response

Response to Reviewer 2 Comments

Dear reviewer:

We are very grateful for your careful review regarding our manuscript entitled “Influence of microwave frequency and gas humidity on the vitro blood coagulation in cold atmospheric pressure plasma” (Manuscript # processes-1338471). We all agree that your comments are highly valuable and constructive for us. According to your comments, we have revised the manuscript thoroughly and carefully. We look forward to hearing from you regarding our submission. We would be glad to respond to any further questions and comments that you may have.

For all your concerns in your comments, below are our detailed responses. Hope both the revised manuscript and the responses to your comments can meet your expectations and requirements.

Thank you very much for your effort in reviewing our work again.

Best regards,

Jie YU, Li WU, and Kama HUANG

Point 1: In the title and everywhere in the text instead of "vitro" have to be written "in vitro" or "in-vitro".

Response 1: Thank you for your suggestion. All the "vitro" in this title and text have been changed into "in-vitro" in the revised manuscript.

Point 2: The quality of the figures and the text inside them is not good and it is difficult to read and understand it. 

  1. 3 rows 85-88: The text is not clear. Language editing is needed. The authors have used "The first group was driven by two types microwave frequency with 2450 MHz and 5800 MHz..." which is not correct phrase. Similar phrase "two frequencies with 2450 MHz and 5800 MHz" is used on p. 7 row 133.
  2. 6 rows 121-122 The authors concludes that "From these figures, it is obvious that the increasing of humidity does not leading to higher densities of OH and O. Instead, the ROS shows a fluctuating trend." This conclusion follows ONLY from Fig. 5(b) about O. In Fig. 5(a) is well visible increasing almost liner trend of the OH relative line intensity with the humidity.

Response 2: Thank you for your comment. The manuscript has been revised thoroughly. All the errors you mentioned have been revised and marked in the revised manuscript.

Round 2

Reviewer 2 Report

Dear authors,

The objectives and conclusions of the paper as well as their bounds are specified more clearly than in the first version.

The text of the paper has been improved and/or additional details have been added where it has been requested.

I understand that the work represents a preliminary and limited study and results are a base for the further developments.

As a result I recommended the paper publication after minor changes related to English language .

I believe that a revision of English used is still needed.
For example, without exhausting possible other places, some changes should be made as:

1) In Abstract:

"Our results propose a method which is through adjusting microwave frequency and gas humidity to accelerate the in-vitro blood coagulation in CAPP, and suggest a clinical benefit plasma treatment as coagulation device in surgery."

I suggest:

"Our results propose a method to accelerate the in-vitro blood coagulation in CAPP by adjusting microwave frequency and gas humidity, and suggest a clinical benefit as coagulation device in surgery."

2) In Materials and Methods:

"The second group was to distinguish between the effects of heat and plasma on blood coagulation."

In literature, the usual term for that is more simply expressed as : "control group".
For example see:https://www.britannica.com/science/control-group

3) In 4.2
"It is also worth noticing that the plasma with higher gas humidity requires shorter treated time to form clot layer on blood samples."

I think that correct form is: "….shorter treatment time….", with " treatment " instead of "treated".

Similar for Conclusion 2

"treatment time" instead of "treated time".

4) Figure 4 caption

"….in the integration time of 30 ms…." replaced by : "….(integration time of 30 ms)…." like in Figure 5 caption.

5) In Conclusion 3

"The clotting effect in this work attributes to the ROS in plasmas."

I suggest:

"The clotting effect in this work is attributed to the ROS in plasmas."

My advice is to recheck, thoroughly and carefully, the English style of the whole paper.

Author Response

Response to Reviewer 2 Comments

Dear reviewer:

We are very grateful for your careful review regarding our manuscript entitled “Influence of microwave frequency and gas humidity on the vitro blood coagulation in cold atmospheric pressure plasma” (Manuscript # processes-1338471). We all agree that your comments are highly valuable and constructive for us. According to your comments, we have revised the manuscript thoroughly and carefully. We look forward to hearing from you regarding our submission. We would be glad to respond to any further questions and comments that you may have.

For all your concerns in your comments, below are our detailed responses. Hope both the revised manuscript and the responses to your comments can meet your expectations and requirements.

Thank you very much for your effort in reviewing our work again.

Best regards,

Jie YU, Li WU, and Kama HUANG

Point 1: In Abstract:

"Our results propose a method which is through adjusting microwave frequency and gas humidity to accelerate the in-vitro blood coagulation in CAPP, and suggest a clinical benefit plasma treatment as coagulation device in surgery."

Response 1: Thank you for your suggestion. It has been changed to “Our results propose a method to accelerate the in-vitro blood coagulation in CAPP by adjusting microwave frequency and gas humidity, and suggest a clinical benefit as coagulation device in surgery.”.

Point 2: In Materials and Methods:

“The second group was to distinguish between the effects of heat and plasma on blood coagulation.”

Response 2: Thank you for your comment. It has been changed to “The control group was to distinguish between the effects of heat and plasma on blood coagulation.”. All the errors you mentioned have been revised in the revised manuscript.

Point 3: In 4.2

"It is also worth noticing that the plasma with higher gas humidity requires shorter treated time to form clot layer on blood samples."

Response 3: Thank you for your comment. It has been changed to “The control group was to distinguish between the effects of heat and plasma on blood coagulation.” All the errors you mentioned have been revised in the revised manuscript.

Point 4: Figure 4 caption

"….in the integration time of 30 ms…." replaced by : "….(integration time of 30 ms)…." like in Figure 5 caption.

Response 4: Thank you for your comment. It has been changed to “Spectrums of plasma plumes under a microwave input power of 20 W (integration time of 30 ms) at different frequencies. (a) 2450 MHz, (b) 5800 MHz.”.

Point 5: In Conclusion 3

"The clotting effect in this work attributes to the ROS in plasmas."

Response 5: Thank you for your comment. It has been changed to “The clotting effect in this work is attributed to the ROS in plasmas.”.
